# The Impact of Air Pollution and Obesity on Cognitive Decline and Risk of Alzheimer’s Disease

**DOI:** 10.3390/ijms27010092

**Published:** 2025-12-21

**Authors:** Zoe A. Keller, Katherine M. Eggers, Joshua P. Nixon, Tammy A. Butterick

**Affiliations:** 1Minneapolis Veterans Affairs Health Care System, Minneapolis, MN 55417, USA; zoe.keller@va.gov (Z.A.K.);; 2Center for Veterans Research and Education, Minneapolis, MN 55417, USA; 3Department of Surgery, University of Minnesota, Minneapolis, MN 55455, USA; 4Department of Neuroscience, University of Minnesota, Minneapolis, MN 55455, USA; 5Department of Food Science and Nutrition, University of Minnesota, St. Paul, MN 55108, USA

**Keywords:** particulate matter, plaque, amyloid, inflammation, oxidative stress, cognitive dysfunction, Wnt signaling pathway, mental health, adiposity, leptin, inflammation

## Abstract

Obesity and air pollution are two pervasive and increasingly prevalent risk factors for neurodegenerative diseases, like Alzheimer’s disease. Both independently disrupt brain homeostasis through overlapping mechanisms, including chronic neuroinflammation, oxidative stress, and insulin resistance. Recent evidence highlights the Wnt/β-catenin signaling pathway as a critical integrator of these insults, mediating neuroprotective processes such as synaptic plasticity, blood–brain barrier integrity, and neuronal survival. In this review, we synthesize emerging data on how obesity-driven metabolic dysfunction and air pollution-induced oxidative injury synergize to impair brain metabolism and accelerate cognitive decline. We describe the roles of pathways such as JAK-STAT, NF-κB, and TLR4 signaling cascades, as well as leptin and adiponectin imbalances, in modulating glial reactivity and neuroimmune signaling. Particular attention is given to the suppression of Wnt/β-catenin signaling in obese and pollution-exposed brains, and its consequences for Alzheimer’s disease pathology, including β-amyloid accumulation and tau hyperphosphorylation. Finally, we examine the translational implications, highlighting the Wnt pathway as a potential therapeutic target that offers neuroprotection in the context of dual metabolic and environmental stress. Together, these insights provide a mechanistic framework that links systemic dysfunction to central nervous system vulnerability, offering pathways for intervention in at-risk populations.

## 1. Introduction

The global prevalence of dementia and associated neurodegenerative disorders continues to rise at an alarming rate, presenting challenges for both clinical management and public health [1]. While aging and genetic factors, such as the apolipoprotein E4 (APOE4) allele, have been recognized as risk factors, there is growing evidence linking modifiable determinants, including obesity and exposure to air pollution, to the development of neurodegenerative diseases such as Alzheimer’s disease (AD) [2,3,4,5,6]. Obesity has been associated with a state of chronic, low-grade systemic inflammation, often referred to as “metainflammation”, caused by adipocyte dysfunction and persistent immune activation [7,8]. Adipocytes and infiltrating immune cells in obese tissue secrete elevated levels of pro-inflammatory cytokines such as tumor necrosis factor-alpha (TNF-α), interleukin-6 (IL-6), and interleukin-1β (IL-1β), as well as saturated fatty acids (SFAs) [7,8,9]. Dysregulation of metabolic hormones, particularly leptin and adiponectin, coupled with insulin resistance and oxidative stress, further exacerbates neuronal vulnerability and impairs the clearance of amyloid-β protein—particularly amyloid-beta (Aβ) 40 and Aβ42 [10,11,12]—correlating metabolic dysfunction with neurodegenerative pathology.

Obesity is the only well-established modifiable risk factor for Alzheimer’s disease (AD) [13]. AD is characterized by a progressive cognitive decline linked to the accumulation of Aβ plaques and neurofibrillary tau tangles in the brain, both of which are believed to be exacerbated by chronic neuroinflammation and oxidative stress [11]. These mechanisms are increasingly recognized as being influenced by both metabolic factors and environmental exposures, such as air pollution [14]. Current evidence suggests that air pollution may also contribute to the progression of AD by promoting neuroinflammatory pathways, oxidative damage, and blood–brain barrier (BBB) disruption [15,16]. If a definitive causal relationship is established, it would emphasize the importance of reducing exposure to air pollution as a strategy to lower AD risks.

Low-income populations are disproportionately affected by both obesity and air pollution, putting them at a higher risk for developing AD [3,17]. Socioeconomic disadvantage is associated with increased rates of obesity, related metabolic disorders, and dementia [17,18,19,20,21], and these same groups are more likely to be exposed to greater amounts of air pollution due to closer proximity to traffic and industrial sources [14,22,23]. Low-income communities frequently have limited access to environments with clean air due to economic constraints and historic housing policies, worsening health disparities related to air pollution exposure [23]. Studies in Mexico City, a highly polluted urban environment have found that children and young adults display earlier neuropathological changes like Aβ accumulation compared to less-exposed rural areas [12,22]. Urban areas in China and India with high levels of pollution also have a significantly higher incidence of cognitive impairment correlated with ambient pollution levels, disproportionately affecting socioeconomically disadvantaged groups [23]. Together, these intersecting risk factors increase vulnerability to neuroinflammation, accelerate neurodegenerative processes, and increase the incidence of AD in these groups.

Particulate matter (PM) is a major air pollutant that can reach the brain through various routes. Fine particulate matter (PM_2.5_), which has a diameter of ≤2.5 μm, has been identified as a potential environmental factor that contributes to dementia risk [2]. This pollutant can breach the central nervous system (CNS) via systemic circulation or directly through the olfactory nerve [12,24]. PM_2.5_ and associated compounds stimulate innate immune receptors like toll-like receptor 4 (TLR4) on resident immune cells, activating the nuclear factor κB (NF-κB) and Janus kinase (JAK)–signal transducer and activator of transcription (STAT) signaling cascades, inducing a pro-inflammatory state [25,26]. Epidemiological studies have identified a link between exposure to air pollution and both increased obesity prevalence and greater risk of cognitive decline or dementia [1,3]. Air pollution exposure is strongly correlated with impaired insulin signaling, increased leptin resistance, and elevated inflammatory markers in the brain, mirroring physiological changes observed in obesity [25,27,28]. 

The Wnt/β-catenin signaling pathway plays a key role in protection against cognitive decline [29]. Central to both metabolic and environmental insults, the Wnt pathway plays key roles in synaptic maintenance, BBB integrity, mitochondrial function, and neuronal survival [29]. Chronic inflammation and oxidative stress have been linked to suppression of the Wnt pathway, which, in turn, leads to the accumulation of amyloid-β protein, tau hyperphosphorylation, and accelerated cognitive decline [9,11,30]. 

This review compiles recent findings on the mechanistic overlap between obesity and air pollution, with a focus on the Wnt/β-catenin signaling pathway and discusses implications for Alzheimer’s disease and related cognitive disorders.

## 2. Materials and Methods

This review draws from peer-reviewed articles, cohort studies, meta-analyses, and animal studies to examine the association and mechanistic overlap between air pollution, obesity, the Wnt/β-catenin signaling pathway, and cognitive decline.

### Search Strategy and Selection Criteria

To compile current evidence connecting obesity, air pollution, Wnt/β-catenin signaling, and cognitive decline, a comprehensive literature search strategy was employed. Peer-reviewed articles, animal studies, cohort studies, and meta-analyses published in English from 2005 to July 2025 were considered. Primary searches were conducted in PubMed (including MEDLINE) and Google Scholar using focused query terms: “PM2.5 obesity air pollution dementia Wnt”, “obesity neuroinflammation air pollution”, “obesity air pollution insulin resistance”, “obesity PM_2.5_ dementia”, “Wnt/β-catenin neuroinflammation Alzheimer’s”, “air pollution adiposity”, “air pollution mental health”, “air pollution blood–brain barrier obesity”, “air pollution HPA”, “air pollution cognitive decline”, “obesity cognitive decline”, “air pollution socioeconomic dementia” and “air pollution stress brain” (Table 1).

We focused on published literature that examined the relationships between obesity, air pollution, neuroinflammation, the Wnt/β-catenin pathway, and cognitive decline. The references of the eligible studies were manually checked (snowballing) to find additional sources not identified in direct database queries. Many search results contained irrelevant or repetitive information and/or studies, which were excluded. Titles and abstracts were read to determine relevance. For articles deemed relevant, full texts were reviewed. Articles that did not appear to be relevant, or where full texts could not be accessed, were excluded from analysis.

Authors also utilized AI search tools (ChatGPT Scholar, Perplexity) to identify newly published or hard-to-find sources. Several key references, including Yu et al. (2024) [3], Dai et al. (2025) [26], and Sharma et al. (2025) [24], were identified through AI-assisted literature searches and cross-validation with conventional literature databases. These sources were double-checked to ensure validity and read in full before being added. AI tools were not used to summarize articles.

## 3. Obesity and Metabolic Dysregulation Induced by PM_2.5_ Exposure

Obesity is linked to chronic, low-grade systemic inflammation resulting from an excess of nutrients and dysfunction of adipocytes [7,8]. In obese tissue, adipocytes and infiltrating immune cells release elevated levels of saturated fatty acids and pro-inflammatory cytokines, including TNF-α, IL-6, and IL-1β [7,8,9].

The JAK/STAT pathway plays a crucial role in the neuroinflammatory response to obesity. In obesity, STAT3 is chronically stimulated due to elevated levels of IL-6 and leptin [27]. This is believed to lead to the overexpression of suppressor of cytokine signaling 3 (SOCS3), which inhibits leptin receptor signaling and contributes to central leptin resistance [27]. Persistent activation of JAK-STAT/SOCS3 is thought to sustain a pro-inflammatory milieu in the hypothalamus, further exacerbating neuronal stress and central insulin resistance [31]. Persistent activation in obesity leads to a disruption in leptin signaling and an increase in central inflammation [9].

Oxidative stress and mitochondrial dysfunction further exacerbate damage, while insulin resistance and adipokine imbalance (characterized by low adiponectin and dysregulated leptin) impair neuronal function and Aβ clearance. Cellular senescence and senescence-associated secretory phenotype (SASP) factors promote inflammation, aging, and cognitive deficits.

### 3.1. Epidemiological Evidence for PM_2.5_-Induced Obesity

Recent large-scale epidemiological analyses provide evidence that air pollution is associated with an increased risk of obesity in adults, even when factors like diet and exercise are considered [3]. Specifically, exposure to PM_2.5_ and particulate matter ≤ 10 μm (PM_10_) is correlated with a higher body-mass index and an increased incidence of obesity, with more pronounced effects in men, older adults, and rural populations [3]. Surprisingly, exposure to the pollutants sulfur dioxide and carbon monoxide (CO)was associated with lower rates of obesity, highlighting the complexity of pollutant-specific effects [3]. It is also important to consider that these relationships are not only direct but are also mediated by behavioral and psychosocial factors such as decreased physical activity, sleep disturbance, and negative impacts on mental health [3]. These behavioral effects may be influenced by air quality, further tying pollution exposure to metabolic dysfunction.

### 3.2. Mechanisms: Air Pollution, Obesity, Inflammation, and Metabolic Dysfunction

Particulate matter such as PM_2.5_ is believed to reach the brain through various routes. One pathway involves transport directly through the olfactory nerve, where inhaled particles pass through the BBB and reach the CNS via the olfactory bulb [24]. This route has been confirmed by studies in which animals were exposed to urban air, demonstrating the accumulation of particulates and oxidative damage primarily in the olfactory bulb and frontal cortex [24]. PM_2.5_ and associated inflammatory mediators may also enter circulation via the lungs and later penetrate the BBB, delivering a secondary injury to the brain parenchyma [24].

Both obesity and air pollution are associated with chronic innate immune activation via overlapping signaling mechanisms [7,9,12]. For example, TLR4 recognizes endogenous danger signals, like SFAs, and exogenous ligands, like lipopolysaccharide (LPS) [7,9,12]. In obesity, an increase in gut permeability enhances LPS translocation into the bloodstream, activating TLR4 on microglia and astrocytes [7,12]. Similarly, PM in air pollution frequently carries LPS and other pathogen-associated molecular pathways that stimulate TLR4 in pulmonary and CNS tissue [9,12]. Both of these are believed to lead to the downstream activation of the myeloid-differentiation primary response 88 (MyD88)-NF-κB pathway, causing a sustained release of pro-inflammatory cytokines like IL-1β, IL-6, and TNF-α [7,13]. The compounding effects of both obesity and PM exposure may result in synergistic or prolonged signaling, ultimately overwhelming endogenous regulatory systems and maintaining a constant state of chronic inflammation [32].

Obesity and cognitive decline are both thought to be exacerbated by exposure to air pollution. Chronic exposure to PM is associated with systemic inflammation, oxidative stress, and metabolic disruptions, which result in increased adiposity and an increased risk of neurodegeneration [33,34,35]. Population studies reveal that chronic exposure to PM_2.5_ may accelerate cognitive decline through various complex pathways centering on leptin resistance [27]. PM_2.5_ can contain LPS, which activates TLR4 signaling in key brain regions involved in energy homeostasis and cognition, including the hippocampus and hypothalamus [27]. This activation promotes NF-κB-mediated production of SOCS3, inhibiting leptin receptor signaling and ultimately contributing to central leptin resistance [27]. Leptin resistance disrupts the satiety and energy balance in the brain, contributing to increased adiposity and obesity [27]. Impaired leptin signaling in the brain is also linked to an increased accumulation of Aβ and tau hyperphosphorylation, both of which are associated with cognitive decline and AD pathology [1,25]. PM_2.5_ exposure also exacerbates neuroinflammation via the gut–brain axis [36]. Systemic inflammation is likely further compounded by alterations in the gut microbiota, which are associated with increased intestinal permeability [27]. These effects are more prevalent in obese individuals, as pre-existing gut dysbiosis and increased intestinal permeability worsen the inflammatory response triggered by PM_2.5_, further contributing to neuroinflammatory signaling and cognitive decline [1,3,25]. Animal studies suggest that exposure to air pollution contributes to alterations in gut microbiota, worsening these effects and negatively affecting metabolic health (Figure 1) [36].

## 4. Neurotoxic Effects of Air Pollution and Impact on Cognitive Decline

Air pollution may cause CNS inflammation in ways that are very similar to obesity-related metainflammation [7,12]. Postmortem human studies revealed an increase in microglial activation, elevated NF-κB in endothelial cells, BBB disruption, and upregulation of inflammatory cytokines and enzymes—including IL-1β and cyclooxygenase-2 (COX-2)—in brain samples from polluted cities [12]. Mouse studies have also found an increased production of COX-2 after exposure to diesel, a known air pollutant, along with alterations in fatty acid composition and membrane fluidity [37,38]. Researchers also found an increase in beta-site amyloid precursor protein cleaving enzyme 1, a protein that produces Aβ [37]. Exposure to large amounts of the pollutants carbon monoxide and nitrogen dioxide (NO_2_) has been linked to increased incidence of dementia in population studies [39,40]. NO_2_ inhalation has also been linked to aggravated Aβ42 accumulation in mice [5].

Toxin components of PM, including PM_2.5_, can bind to metals or bacterial endotoxins, stimulating innate immune receptors on microglia (such as TLR4) and leading to myeloid differentiation primary response 88 (MyD88)-dependent activation of NF-κB [12,25]. This causes a cascade of inflammatory signaling, including NF-κB activation, similar to diet-induced neuroinflammation [7,25]. Experimental animal models exposed to air pollution exhibit elevated levels of pro-inflammatory cytokines in the brain (TNF-α, IL-6), increased NF-κB activity, and increased activation of mitogen-activated protein kinases (MAPKs) like c-Jun N-terminal kinase (JNK) [12,25,41]. These changes are linked to microglial hypertrophy and reactive astrogliosis—two hallmarks of neuroinflammation [12]. Evidence also supports the idea that pollution can activate the JAK-STAT pathway [25,26]. LPS is thought to initiate SOCS3 expression, disrupting leptin and insulin signaling in the brain and further paralleling obesity-related mechanisms [27].

Pollution-induced oxidative stress is believed to disrupt various brain functions. PM_2.5_ often contains transition metals and organic compounds that catalyze the synthesis of hydroxyl radicals and superoxide. These free radicals can damage lipids, proteins, and nucleic acids in the brain [11,25,42]. In a study comparing urban vs. rural animals, researchers found that dogs from polluted cities exhibited significantly higher oxidative DNA damage in the frontal cortex and hippocampus [12]. Animal studies reveal that chronic exposure to pollutants also depletes antioxidants, such as glutathione [11,25]. This contributes to impaired mitochondrial function, causing energy deficits and an increase in neuronal apoptosis. 

BBB integrity is also believed to be affected by exposure to pollution. Vascular inflammation and oxidative stress can damage tight junction proteins in the brain’s endothelial cells [12]. In clinical findings using autopsy data from polluted environments, researchers found elevated levels of albumin leakage into brain tissue, endothelial activation, and immune cell infiltration [12]. Collectively, these observations are consistent with a compromised BBB. These changes are exacerbated when air pollution exposure occurs in conjunction with obesity, compounding their effects and increasing BBB permeability, which allows for greater entry of peripheral cytokines and particulate toxins [13].

Oxidative stress is a major contributor to cognitive decline in both obesity and air pollution exposure. Excessive ROS production, resulting from mitochondrial dysfunction and environmental oxidants, can cause damage to lipids, proteins, and nucleic acids in neural tissue [11,43]. Chronic exposure to PM can increase ROS production through metal-catalyzed reactions and mitochondrial disruption, amplifying the effects of oxidative injury in vulnerable brain regions such as the hippocampus and frontal cortex [12,14]. Elevated oxidative stress can also activate redox-sensitive transcription factors, such as NF-κB and activator protein 1, which activate pro-inflammatory cytokine cascades that further compromise the BBB and promote neuronal apoptosis [25,43].

Air pollution-induced oxidative stress appears to overlap with mechanisms associated with AD pathology. Increased ROS production results in increased Aβ production, increased β-site amyloid precursor protein cleaving enzyme-1 production, and increased tau hyperphosphorylation via activation of JNK and p38 MAP kinases [11,43]. In AD, oxidative damage to synaptic and mitochondrial proteins may impair neuronal metabolism and promote neurodegeneration [11]. Postmortem animal studies suggest that antioxidant depletion and persistent oxidative stress in the presence of PM_2.5_ contribute to synaptic loss and early Aβ deposition [12]. These findings suggest that oxidative stress acts as a mechanistic amplifier on the effects of air pollution, accelerating mitochondrial damage and compounding with metabolic dysfunction to drive AD-like pathology.

### Impact on Brain Health

Chronic exposure to airborne pollutants has been linked to the early onset of neuropathological changes associated with Alzheimer’s and Parkinson’s disease [1,11,12]. For example, one comprehensive study found that dogs raised in areas with pollution develop diffuse Aβ plaques earlier than dogs raised in clean environments [12]. Similar observations have been made in children and young adults from urban areas with high PM exposure [22]. They were found to have early-stage accumulation of Aβ42 and α-synuclein aggregates, as well as signs of increased glial activation and neurovascular damage [22,24]. Human cohort studies also found that populations exposed to high amounts of the pollutant carbon monoxide had an increased incidence of dementia [39]. The United States EPA recognizes various air pollutants, including PM, as harmful to the human body [44]. Many of these pollutants are linked with increased oxidative stress, neuroinflammation, blood–brain barrier disruption, and neuronal damage, all of which are believed to contribute to the progression of neurodegenerative diseases like AD. Studies suggest that long-term exposure to pollutants like ozone and PM_2.5_ above the US EPA standards is associated with an increased incidence of AD [6]. Oxidative stress resulting in Aβ42 overproduction and accumulation in hippocampal tissue has also been linked to ozone exposure in rat studies [45].

Epidemiological evidence also increasingly supports air pollution as a risk factor for dementia. Large-scale studies have shown a correlation between long-term PM_2.5_ exposure, accelerated cognitive decline, and AD pathology [1,3]. These correlations remain even after adjusting for other risk factors. Evidence supports that exposure to pollution may exacerbate or accelerate the progression of dementia, as the upregulation of pro-inflammatory cytokines, oxidative stress, and BBB breakdown are all driven by exposure to pollution and can drive amyloid plaque formation, tau hyperphosphorylation, and impair microglial clearance [11,30].

Air pollution is also thought to contribute to cerebrovascular pathology, increasing the potential risk of vascular dementia [1,3,12]. PM exposure has been linked to small vessel disease, white matter hyperintensities, and microinfarcts [1]. These vascular effects are especially damaging to individuals with cardiometabolic conditions, which are more common in obesity [1,7]. Systemic inflammation resulting from air pollution exposure also worsens alterations in gut microbiota caused by obesity [27,36]. These compounding effects of obesity and pollution accelerate cognitive deterioration. These findings suggest a synergistic effect, with obesity amplifying the neurotoxic effects of air pollution by increasing BBB permeability and inflammation [7,12,14].

## 5. Impact on Mental Health

Both air pollution and obesity are believed to be risk factors for mental health disorders, with epidemiological evidence showing an increase in the incidence of depression, anxiety, and cognitive impairment across populations [18,46]. Findings suggest these risk factors may exhibit a synergistic effect through overlapping mechanisms, but further research is needed to confirm [18,47,48].

### 5.1. Mental Health and Air Pollution

Air pollution has been associated with increased risk of depression, anxiety, and other mental health disorders [47,48]. Large meta-analyses and longitudinal cohort studies reveal links between airborne pollutants (including PM_2.5_, nitrogen oxides, and polycyclic aromatic hydrocarbons) and higher rates of anxiety and depressive symptoms across populations and age groups [47,48]. Exposure to air pollution is linked to measurable reductions in gray matter volume, changes in white matter, and neurostructural changes in important regions, including the hippocampus, prefrontal cortex, and amygdala [48].

Airborne PM can enter the brain directly via the olfactory bulb or across the BBB, resulting in local and systemic inflammation [12,48]. Animal models show that exposure is linked to increased microglial activation, elevated levels of pro-inflammatory cytokines, and higher oxidative stress in frontolimbic regions that are crucial to the development of anxiety and depression psychopathology [48]. Other forms of air pollution (including diesel exhaust particles (DEP) and ozone) induce disruptions in mitochondrial function, lipid peroxidation, and altered astrocyte morphology, further impairing neuronal homeostasis [48]. Studies in both humans and animals demonstrate significant changes in neurotransmitter receptor gene expression, including those for glutamate, gamma-aminobutyric acid (GABA), and dopamine, in groups exposed to air pollution [48]. Altered turnover and metabolism of neuromodulators have also been observed in exposed groups [48]. These changes impact synaptic plasticity and neural circuitry in areas involved in emotional regulation and stress response [48]. Evidence suggests that pollution-related changes in these monoamine systems may partly explain the increased incidence of mental health disorders in groups exposed to air pollution [48].

Individuals exposed to air pollutants during critical developmental periods, such as prenatal, early-life, and adolescence, appear to be especially vulnerable to its effects [48]. Early exposure is linked with an increased risk of developing mental health disorders later in life [48]. There is a significant association found between short-term PM_10_ exposure and suicide risk, including both attempted and completed suicide attempts [47].

### 5.2. Obesity and Mental Health

Epidemiological studies have established a link between obesity and increased rates of mental illness, including depression, anxiety, and related disorders. Meta-analyses and large-scale cohort investigations indicate bidirectionality [18]. Obesity increases the risk of developing depression and related disorders, and depression increases the risk of developing obesity, particularly in adult women [18,46].

Mechanistically, chronic low-grade inflammation is a key feature linking obesity and mental illness. The expansion of adipose tissue caused by obesity is linked to an increased production of pro-inflammatory cytokines (including TNF-α and IL-6), impairing BBB function, increasing microglia activation, and elevating CNS inflammation [7,13,49]. These inflammatory mediators disrupt the neuroendocrine axis, particularly the Hypothalamic–Pituitary–Adrenal axis, causing imbalances in cortisol and promoting both mood disorders and metabolic dysfunction [49]. Adipokine dysregulation, characterized by an imbalance in leptin and adiponectin, further impairs neuronal signaling and plasticity, contributing to depressive symptoms [10,49]. Inflammation is also associated with changes in the metabolism of tryptophan (TRP), which is a precursor of serotonin (5-HT), as pro-inflammatory cytokines induce indoleamine-2,3-dioxygenase, which catabolizes TRP along the kynurenine (KYN) pathway [50]. KYN pathway activation is documented in obese subjects along with a reduction of 5-HT plasma levels, contributing to depressive symptoms [50]. Activation of the KYN pathways leads to the metabolism of kynurenic acid (KYNA), which might be harmful at higher concentrations, causing cognitive impairments and pathophysiology associated with schizophrenia [51].

Population studies have documented a higher incidence of depression and related mental illness in children, adolescents, and young adults, with females being particularly vulnerable [52,53]. Similar epidemiological trends are observed across adult populations worldwide [18,46]. Schizophrenic patients have higher rates of diabetes mellitus than the general population and higher dosages of antipsychotic medication were associated with an increased risk for diabetes [54]. Some nonpharmacological factors of weight gain can be seen in bipolar disorder patients who are reported to have atypical depression symptoms, including eating habits and behavior, physical inactivity, and a lowering of metabolic rate during depressive episodes, all of which contribute to the onset and maintenance of obesity [55,56]. Depression was found to be a risk factor for people with schizophrenia, as not only did depressed mood result in reduced physical activity, increased food intake, and insufficient sleep, but depression also increases the secretion of cortisol by stimulating the hypothalamic–pituitary–adrenal (HPA) axis [57]. This release of cortisol induces weight gain through the accumulation of visceral fat and by inhibiting lipid mobilization [57].

### 5.3. Air Pollution, Stress, and the Hypothalamic–Pituitary–Adrenal (HPA) Axis

Current evidence suggests air pollution exposure can disrupt the HPA axis. The HPA axis is responsible for controlling stress responsiveness through the release of corticotropin-releasing hormone from the hypothalamus, adrenocorticotropic hormone from the pituitary, and glucocorticoids from the adrenal cortex [38]. Persistent activation of the HPA axis can be caused by environmental or physiological stress, resulting in an increase in systemic cortisol, which can impair hippocampal feedback signaling, reduce neurogenesis, and exacerbate neuroinflammation [38,58]. Long-term glucocorticoid elevation has been shown to disrupt neuronal metabolism, increase ROS production, and promote dendritic retraction in stress-sensitive brain regions, including the hippocampus, prefrontal cortex, and amygdala [38,59].

Exposure to air pollution induces psychosocial and biochemical stress responses that trigger chronic HPA activation [22,24]. Experimental data suggest that exposure to fine particulate matter increases hypothalamic inflammatory signaling, resulting in elevated levels of corticosterone in animal models [25]. Chronic activation of the HPA axis increases microglial sensitivity to secondary insults and promotes neuroinflammation [38]. Dysregulated HPA function also impacts leptin and insulin resistance pathways, further impairing energy balance and exacerbating cognitive deficits in obesity and air pollution exposure-linked conditions [13,27]. The compounding effects of air pollution–induced stress, elevated glucocorticoids, and inflammation result in oxidative and neuroendocrine dysfunction that accelerates brain aging and cognitive decline [38]. Integrating HPA dysfunction into models of air pollution exposure could help provide a more comprehensive understanding of the interplay between environmental stressors, neuroendocrine regulation, and neuroinflammatory pathways in the development of cognitive decline [38].

## 6. Role of Wnt/β-Catenin Signaling in Air Pollution and Obesity-Induced Neurodegeneration

The Wnt/β-catenin pathway plays a key role in maintaining brain health by supporting neurodevelopment, synaptic function, and BBB stability [29]. Wnt signaling also promotes β-catenin stabilization and nuclear translocation, where it induces the expression of genes that strengthen synaptic plasticity, enhance mitochondrial resilience, and inhibit the accumulation of toxic proteins like amyloid-β [29]. Recent evidence suggests this pathway is disrupted by chronic inflammation and oxidative stress—both of which can occur as a result of obesity or exposure to air pollution [24]. Pro-inflammatory cytokines such as TNF-α, IL-6, and IL-1β can increase endogenous inhibitors of Wnt signaling [29]. Oxidative damage is thought to destabilize β-catenin, disrupting its neuroprotective effect in the brain [29]. Because of this, the neuroprotective features of Wnt signaling are believed to be impaired in individuals with metabolic and environmental stressors, accelerating neurodegenerative damage [29].

### Therapeutic Potential of Targeting Wnt Signaling

Emerging evidence supports the Wnt pathway’s role in protection against cognitive decline, resulting in a growing interest in Wnt signaling as a therapeutic target against Alzheimer’s disease and pollution-related cognitive decline [29,60]. In animal models of AD, restoring Wnt/β-catenin signaling is shown to enhance neurogenesis, improve synaptic strength, and protect against BBB degradation [30,61]. Restoring impaired Wnt activity is also linked to lower Aβ accumulation and reduced tau phosphorylation [60]. Researchers believe it does this partly by inhibiting glycogen synthase kinase-3β (GSK-3β), a kinase linked to insulin resistance and metabolic dysfunction [29]. These findings suggest restoring the Wnt pathway could help slow the progression of AD pathology or cognitive decline, potentially caused by obesity and air pollution exposure [30].

Pharmacological therapies could enhance Wnt signaling. GSK-3β inhibitors or antibodies targeting Dickkopf-1 (DKK1) have shown promise in preclinical models; however, potential therapies need to be tested rigorously due to potential risks [29]. Tumorigenesis, fibrosis, bone overgrowth, and immune dysregulation are all possible risks of systemic Wnt activation due to its broad regulatory influence [29]. Because of this, approaches that selectively target the brain or modulate pathway activity in a controlled way are particularly of interest to researchers. The role of Wnt signaling in immune cells is also not fully understood, emphasizing the need for further research to fill knowledge gaps and determine when and where Wnt signaling is most protective [29].

Recent studies have shown that, in the context of obesity, genetic modulation of lipid metabolism may impact the Wnt/β-catenin pathway [9]. In a fatty acid binding protein 4 (FABP4)-knockout (KO) model, our laboratory has shown that mice fed a high-fat diet (HFD) exhibited resistance to hippocampal neuroinflammation and cognitive impairment [9,62]. Transcriptomic profiling of the hippocampus revealed increased expression of Wnt/β-catenin signaling components, including β-catenin and disheveled segment polarity protein 3 (Dvl3) [9]. Other studies have shown that the knockout of β-catenin in the hypothalamus was associated with an increased body weight and a reduction in sensitivity to both leptin and insulin [63]. Our data has also shown a reduction in SRY-box transcription factor 6 (Sox6), a known inhibitor of Wnt activity [9,64]. Compared to wild-type controls, FABP4-KO mice maintained synaptic integrity and showed increased memory performance despite exposure to metabolic stress [9]. These findings support the idea that enhanced Wnt/β-catenin signaling supports resilience against obesity-related cognitive decline [30]. Restoring activity in the Wnt/β-catenin signaling cascade shows promise in models of metabolic stress and neurodegeneration [29]. Stabilizing β-catenin or inhibiting its negative regulators (like GSK-3β) could enhance mechanisms for synaptic repair and help preserve BBB integrity.

## 7. Integration of Mechanisms: Interplay Between Air Pollution Exposure, Obesity, Inflammation, and Wnt Signaling in AD Risk

Chronic exposure to air pollution is believed to cause systemic inflammation and metabolic disruptions, such as leptin resistance, resulting in increased adiposity and promoting cognitive decline [2,25]. PM_2.5_ exposure activates TLR4/NF-κB signaling in brain regions such as the hippocampus and hypothalamus, increasing SOCS3 and impairing leptin receptor signaling [27]. Impaired leptin signaling, which is a hallmark of obesity [27], is linked to an increase in amyloid-β accumulation and tau hyperphosphorylation, both of which are associated with AD pathology [27]. Obesity-induced gut microbiota alterations and increased intestinal permeability exacerbate these metabolic and inflammatory impacts, and these effects converge to suppress the neuroprotective Wnt/β-catenin pathway [27].

Exposure to air pollution is believed to directly compromise brain function through microglial activation, BBB disruption, and oxidative damage in neural tissue [1,25]. These insult pathways impair neuronal viability, synaptic plasticity, and promote the accumulation of amyloid-β and tau pathology—all of which are associated with cognitive decline and dementia [1,25]. Epidemiological data support this relationship, as populations with increased exposure to air pollution also have increased rates of obesity and cognitive decline, particularly among those with a genetic predisposition to metabolic dysfunction [2]. This interaction also involves the suppression of neuroprotective pathways, such as the Wnt/β-catenin signaling pathway. While chronic inflammation and oxidative stress can disrupt Wnt signaling, our experimental data in rodent models suggest that the upregulation of specific Wnt ligands, such as Wnt3a, may also exacerbate neuroinflammation and degeneration when overactivated after particulate exposure [65]. We found Wnt3a had a positive log fold change in rat brains after exposure to carbon black naphthalene via inhalation [65]. PM_2.5_ exposure causes chronic inflammation and oxidative stress, which is believed to disrupt Wnt signaling and lead to compromised synaptic repair, reduced neurogenesis, impaired BBB integrity, and increased vulnerability to the metabolic and neurodegenerative consequences of environmental stress (Figure 2) [25]. These findings suggest that air pollution amplifies both metabolic and cognitive vulnerabilities through the convergence of inflammatory, oxidative, and neurological processes [1,2,25].

Long-term inflammation in the brain has been recognized as a key contributor to neurodegeneration, particularly in conditions like AD [13,31]. When pro-inflammatory cytokines like IL-1β and TNF-α are continuously elevated, they create a feedback loop such that ongoing damage stimulates more immune activity [7,31]. In AD, this is observed when microglia surround amyloid plaques and release pro-inflammatory cytokines, reactive oxygen species (ROS), and proteases that exacerbate tau phosphorylation and disrupt synaptic signaling [11]. Obesity appears to accelerate this process, introducing low-grade inflammation earlier in life and increasing the risk for cognitive decline [7,13]. Exposure to air pollution is thought to further exacerbate this, adding an additional layer of inflammatory signaling [12]. 

Both obesity and exposure to air pollution have been associated with an increase in Aβ in the brain [12]. Imaging and autopsies have revealed that humans with obesity during midlife have more extensive amyloid plaques in regions involved in memory, including the hippocampus, parietal, temporal, and frontal cortices, hallmark sites of AD pathology [10]. This is partially because elevated insulin levels compete for the enzyme responsible for breaking down amyloid plaques, insulin-degrading enzyme (IDE) [10,66]. Air pollution also appears to accelerate the formation of amyloid plaques. In animal models, DEPs are believed to lead to a buildup of Aβ [16]. In humans, individuals in high-pollution areas show an increase in Aβ deposits and glial activation [12]. Metabolic and oxidative stress also aggravate tau pathology by activating enzymes that promote tau phosphorylation, such as JNK and p38 MAPK [11].

### Translational Implications and Future Directions

Further investigation is needed to identify exactly how pathways triggered by obesity and pollution interact with the brain, particularly how they converge on shared inflammatory and oxidative stress pathways. While it is believed that both stressors activate immune signaling cascades like the TLR4-MyD88-NF κB signaling axis [7,13], it is still unclear whether their combined effect is additive or synergistic. This distinction could have implications for the severity of neuroinflammatory outcomes. Animal models that investigate the effects of combined exposures, such as HFD and chronic air pollution, are essential for understanding the joint impact these stressors have on microglia activation, BBB integrity, and cognitive decline [24]. Further investigation into FABP4 as a potential therapeutic target could lead to a more effective treatment against obesity-induced brain injury [9]. Overall, clarifying the exact mechanistic overlaps is key to developing a targeted treatment plan that addresses both metabolic and environmental effects on brain injury [31]. 

## 8. Discussion

Emerging evidence suggests that exposure to air pollution, particularly PM_2.5_, triggers neuroinflammatory and oxidative pathways in the brain and has been associated with the development of obesity [2,3]. Large-scale epidemiological studies reveal links between chronic exposure to PM_2.5_ and an increased incidence of obesity, even after taking behavioral and genetic factors into account [3]. Mechanistically, exposure to air pollution and obesity are believed to have similar effects on the body, including systemic inflammation and disruptions in metabolic hormone signaling, such as leptin resistance and altered glucose metabolism [25,27]. Animal models also demonstrate that exposure to PM_2.5_ can impair hypothalamic function, increase adiposity, and enhance susceptibility to metabolic dysregulation, further highlighting air pollution’s role as a driver of obesity and cognitive decline (Table 2) [25]. Obesity causes similar effects, and obesity linked to air pollution exposure may further increase the risk of cognitive decline by amplifying chronic inflammation, oxidative stress, and BBB disruption (Table 3) [7,12]. Both risk factors converge on multiple key signaling pathways that impair Wnt/β-catenin activity, such as NF-κB, JAK-STAT, and TLR4 [9,29]. 

Suppression of Wnt/β-catenin signaling accelerated amyloid-β accumulation, tau phosphorylation, and synaptic dysfunction, all of which have been associated with AD pathology and cognitive decline [11,30]. These findings underscore the importance of addressing air pollution as a modifiable environmental risk factor for brain health. It is essential to not only reduce direct neurotoxic effects but also to mitigate the rising frequency of obesity and its impact on brain health. Public health interventions focused on improving air quality may offer benefits, including decreased rates of obesity and protection against cognitive decline [1,3]. Translational strategies—such as targeting Wnt/β-catenin signaling, metabolic regulation, and anti-inflammatory therapies—require further investigation, but could potentially help individuals exposed to metabolic and environmental stress [9,30].

### Limitations

This review has several inherent limitations. First, the literature search was conducted primarily using Google Scholar and PubMed, meaning that potentially relevant sources not in these databases were missing. Second, our exclusion criteria restricted studies to those published in English between 2005 and July 2025, which excluded relevant works written in other languages or published outside this timeframe. Third, only the titles and abstracts were used to determine whether a source was relevant, meaning that texts with useful information buried in the main text were not utilized. Fourth, as a literature review, this work aimed to synthesize and summarize existing research instead of following the strict structured rules of a systematic review, which utilizes predefined protocols, detailed quality appraisal of studies, and a formal assessment of potential biases. While key mechanistic relationships are highlighted in this review based on trends in the literature, these represent associations instead of direct experimental evidence of causality.

Finally, most studies supporting claims about obesity, air pollution, and cognitive impairment are observational, cross-sectional, or animal model-based, which limits the ability to confidently apply findings to clinical populations or infer causality. More systematic reviews and prospective cohort studies with biomarker assessments and standardized exposure protocols are needed to build on these findings and mechanistically clarify the links described here. Despite these limitations, literature reviews are valuable for integrating multiple pieces of evidence and providing guidance for further research into the complex interplay between metabolic and environmental stressors on brain health.

## Figures and Tables

**Figure 1 ijms-27-00092-f001:**
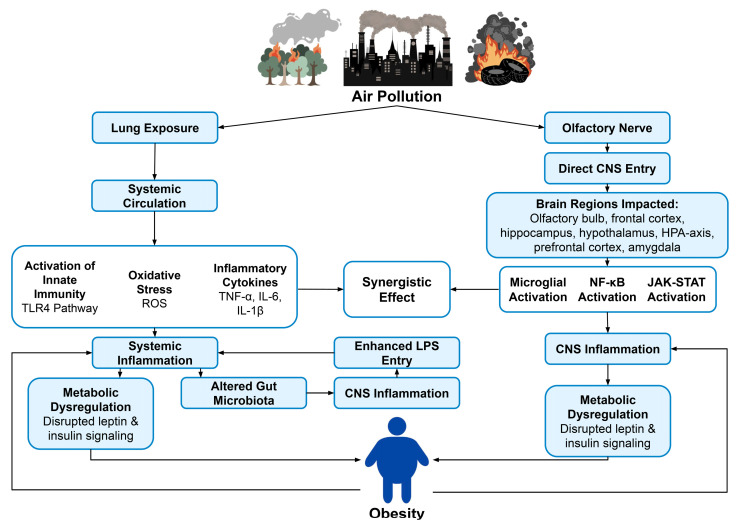
Mechanisms linking air pollution and obesity.

**Figure 2 ijms-27-00092-f002:**
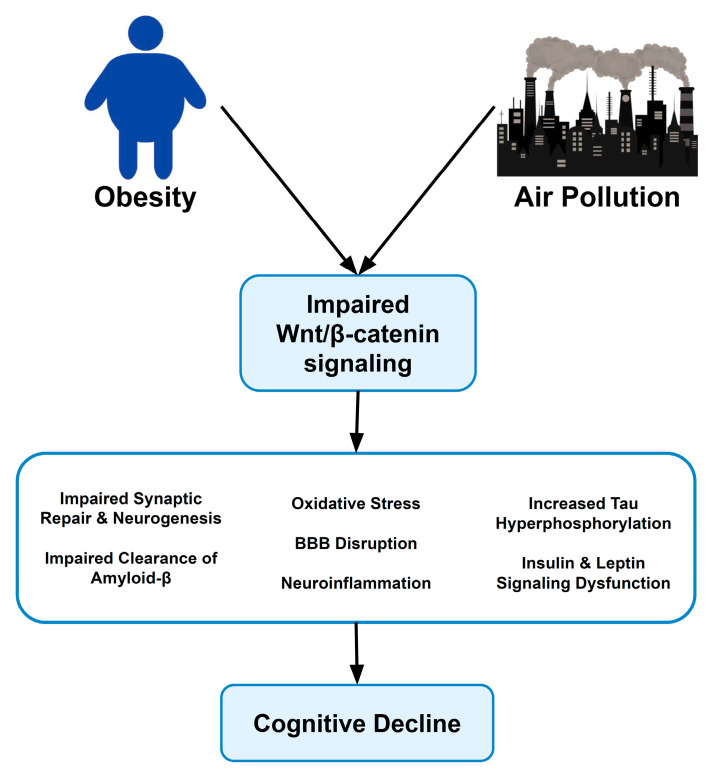
Obesity and air pollution converge on impaired Wnt/β-catenin signaling.

**Table 1 ijms-27-00092-t001:** Inclusion and exclusion criteria table.

Category	Inclusion Criteria	Exclusion Criteria
Study Types	Peer-reviewed research Cohort, case–control, meta-analyses Systematic/narrative reviews	Editorials, letters Abstracts only Case reports/series only
Date Published	2005–July 2025	Published before 2005 or after July 2025
Language	Literature published in English	Literature published in non-English languages
Search Results	Direct database hits (PubMed) Google Scholar Cross-references/snowballing AI-enabled tools (ChatGPT Scholar, Perplexity)	Not captured in these tools
Content	Relating to mechanisms linking obesity, air pollution, neuroinflammation, Wnt/β-catenin, and cognitive decline, general background info	Irrelevant topics

**Table 2 ijms-27-00092-t002:** Summary of Models Investigating Air Pollution and Cognitive Decline.

Species	Pollutant Type & Exposure Route	Exposure Duration	Brain Regions Assessed	Major Endpoints	Key Findings	Mechanistic Insights or Targets	Reference
Dogs, humans	Urban PM (ambient exposure)	Chonic (yr.)	Frontal cortex, hippocampus	Neuroinflammation (cytokines), MRIs, psychometric testing	Aβ and α-synuclein accumulation, cognitive deficits	Chronic oxidative stress, microglial activation, BBB disruption	[12]
Rat (Fischer 334)	Diesel exhaust particles, inhalation	6 h/d, 7 d/w for 6mos.	Cortex, hippocampus	Cognitive testing (memory, learning tasks)	↑ IL-6, TNF-α, early neurodegenerative markers	NF-κB activation, oxidative stress, lipid peroxidation	[16]
Mouse (BALB/c)	Urban PM, whole body exposure, ovalbumin, intranasal instillation	4 h/d, 5 d/w for 2 wks, daily intranasal instillation	Whole brain	Neuroinflammation markers measured	↑ inflammatory markers	↑ TNF-α, oxidative stress in brain tissue	[28]
Mouse (C57BL/6)	Ozone + polystyrene nanoplastics, inhalation + oral gavage	30 d	Prefrontal cortex	Behavioral assays (cognitive and anxiety-like behavior)	Cognitive impairment, anxiety-like behavior	Pyroptosis, mitochondrial dysfunction, oxidative stress	[26]
Mouse (C57BL/6, TLR4-deficient)	PM_2.5_, chamber exposure	1 h/d for 11 wks	Hippocampus, hypothalamus	Cytokine assays, leptin & SOCS3 expression	Inflammation in hypothalamus, leptin resistance,	TLR4/IκBKE activation, JAK-STAT dysregulation	[25]
Human	Chronic PM_2.5_ exposure (population)	Long-term (yr.)	N/A (endothelial and plasma biomarkers)	Circulating endothelial markers, cytokines	Vascular inflammation, endothelial injury	NF-κB activation, systemic inflammation, impaired BBB	[41]
Mouse, HT22 cells	Diesel exhaust particles	24 h exposure	In vitro neuronal cell cultures	Lipidomics, oxidative stress markers	Mitochondrial stress and lipid remodeling	Lipid peroxidation, oxidative stress	[37]
Mouse (C57BL/6J, p47^phox−/−^)	Early-life PM_2.5_ exposure, whole body inhalation, high-fat diet	Developmental–6 h/d, 5 d/w, for 10 wks	Whole brain	Body composition, cytokines, metabolic markers	↑ adiposity and inflammation in adulthood	P47^phox^-oxidase driven oxidative stress	[34]
Mouse (C57BL/6, CCR2^−/−^)	PM_2.5_ exposure, inhalation, high-fat diet	6 h/d, 5 d/wk for 17 wks	Hippocampus, hypothalamus	Glucose tolerance, insulin signaling, cytokines	Insulin resistance, neuroinflammation	CCR2-mediated inflammatory pathway	[35]
Rat (Sprague Dawley)	Carbon black + naphthalene inhalation	6 h/d for 3 d	Whole brain	RNA-seq, cytokine assays	Transcriptomic evidence of neuroinflammation	Wnt/β-catenin dysregulation, oxidative stress	[65]

**Table 3 ijms-27-00092-t003:** Summary of Models Investigating Obesity and Cognitive Decline.

Species	Induction Method	Duration	Brain Regions Assessed	Major Endpoints	Key Findings	Mechanistic Insights or Targets	Reference
Mouse (FABP4 knockout)	High-fat diet	12 wks	Hippocampus	RNAseq, Y-maze	FABP4-KO mice resistant to neuroinflammation and cognitive decline	Upregulated Wnt/β-catenin signaling, reduced Sox6	[9]
Mouse (db/db) (leptin receptor deficient)	Genetic obesity	14–20 wks	Hippocampus	Immunostaining for Aβ and tau, cognitive tests	↑ Aβ plaques and tau phosphorylation	Brain insulin resistance, oxidative stress	[66]
Mouse (C57BL/6, CCR2^−/−^)	PM_2.5_ exposure, inhalation, high-fat diet	6 h/d, 5 d/w, 17 wks	Hippocampus, hypothalamus	Glucose tolerance, insulin signaling, cytokines	Insulin resistance, neuroinflammation	CCR2-mediated inflammatory pathway	[35]
Mouse (C57BL/6J, p47^phox−/−^)	Early-life PM_2.5_ exposure, whole body inhalation, high-fat diet	Developmental, 6 h/d, 5 d/w, for 10 wks	Whole brain	Body composition, cytokines, metabolic markers	↑ adiposity and inflammation in adulthood	P47^phox^-oxidase driven oxidative stress	[34]

## Data Availability

No new data were created or analyzed in this study. Data sharing is not applicable to this article.

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
