# Peer review of "The Impact of Air Pollution and Obesity on Cognitive Decline and Risk of Alzheimer’s Disease"

_ijms, 2025, doi:10.3390/ijms27010092_

Round 1
Reviewer 1 Report
Comments and Suggestions for Authors
ijms-3946106
The Impact of Air Pollution and Obesity on Cognitive Decline 2 and Alzheimer’s Disease Risk
The authors have compiled a review regarding the combination of air pollution and obesity on cognitive decline and Alzheimer’s disease. It is well written and addresses an important topic for human health. There some suggestions for edits and concerns listed below.
Minor concerns/Edits
line 39
“linking modifiable 38 determinants, including obesity and exposure to air pollution” ..to neurodegenerative disorders or Alzheimer's specifically? Please complete the sentence
Line 53-54 This sentence requires a reference.
Line 66 cascades, and inducing a pro-inflammatory state [16,17].
Line 69 Pollution should not be capitalized.
Line 72 This sentence needs a reference.
Line 91-94 The statement of purpose ‘to examine the association and mechanistic overlap between air pollution, 84 obesity, the Wnt/β-catenin signaling pathway, and cognitive decline.”
Includes the term ‘cognitive decline’ but it was not used as search term in the list.
Line 122 It would be best to make a definitive sentence based on the data in the manuscript than to state something is likely to happen.
Line 159-160 Is this all hypothesis or is there some research evidence to describe in this case? Best to rely on the original research results – that is what you are reviewing.
Line 164 – ‘likely’
Line 187 “of inflammatory cytokines— including IL-1β and cyclooxygenase-2 (COX-2)” COX-2 is and enzyme it is not a inflammatory cytokine. Please edit to be factually correct.
Line 190 “likely leading” the manuscript has an overuse of this term ‘likely’ on its own, and then it does not give the concrete data that the supposition is based on.
Line 198 Endotoxin was already addressed in line 152 as lipopolysaccharide – it would be better to have the whole manuscript be consistent.
Line 215 “likely’
Line 216 ‘potentially’
Less theory and more demonstrated results would be a better manuscript. At minimum enough rewording to not have this all be supposition.
Line 265 “ozone, induce disruptions in mitochondrial” please edit the comma.
Line 278 PM10 was not introduced or defined and should be because it differs from PM2.5– it can be confusing to readers unfamiliar with the terms.
Line 311 Reference #30 is review paper from 2011 with no significant follow up or citations. The comments in sentences Lines 310-313 should cite original research rather than review – even if it is the original research that has been cited in review (#30). There should be an inclusion of more current research on this topic especially in light of the use of GLP-1 inhibitors for diabetes and weight loss that may be altering the landscape of these correlations.
Line 326 ‘likely’
Line 328-346 The first part of section 6.1 relies mostly on reference #20 which is from 2014, it would be best to update this section to include more and more recent literature on the topic.
Lines 348-357 This is entire section is based on self-citation of the authors -
Line 406 “in regions involved in memory” this statement is overly vague, it would be better to note the specific brain regions indicated.
Line 407 “likely”
Line 431 formatting needed for PM2.5
Line 439 formatting needed for PM2.5
Line 443 ‘likely’
Line 454 investigation, but could potentially add comma
Major concerns
Line 55-56
‘and these same groups are more likely to be exposed to greater amounts of air pollution due to closer proximity to traffic and industrial sources [13,14].’
Reference 13 is a review paper that speaks to an increased burden on the brains of urban children but does not further detail this into socioeconomic groups or specific location. Reference 14 describes the mechanisms of air pollution causing neuroinflammation and disease.
While it may be that location matters, neither of these references demonstrate this point. Also, this reviewer finds it important to refer to the original research that demonstrated these ideas, especially for review papers.
Because these points are the premise of the review paper they should have appropriate references and include the relevant data on location if that is what is claimed. Please add the required references or restate the sentences to adhere to the data in the literature.
Line 50
“while current evidence suggests that air pollution may also contribute to the development of AD [10]”
The reference #10 [ Miller, A.A.; Spencer, S.J. Obesity and Neuroinflammation: A Pathway to Cognitive Impairment. Brain. Behav. Immun. 2014, 42, 519 10–21, doi:10.1016/j.bbi.2014.04.001] is a review paper, would not be considered current as it is now over a decade old, and moreover does not address air pollution.
This is the premise of the review and yet there is only one, nonapplicable reference to the background of air pollution contributing to AD. This needs to be addressed with recent appropriate publications to substantiate the background for this work.
No where in the paper is the measure of the air pollutions levels in the polluted environment or the control ‘clean’ rural environment described. There is also no mention of any national or state (if applicable) thresholds in place for health and safety.
Author Response
I. Minor Concerns
Comment (Lines 39–40):
Please clarify whether “modifiable determinants” refers to neurodegenerative disorders or Alzheimer’s disease specifically.
Response:
We revised the sentence to specify that these determinants are linked to the development of neurodegenerative diseases such as Alzheimer’s disease (lines 38–41).
Comment (Lines 53–54):This sentence requires a reference.
Response: A supporting reference was added to substantiate this statement (line 57).
Comment (Line 187): COX-2 is an enzyme, not a cytokine.
Response: The text now correctly identifies COX-2 as an inflammation-associated enzyme (line 207).
Comment (Line 278): Define PM₁₀ and distinguish it from PM₂․₅.
Response: PM₁₀ was defined explicitly in the text and added to the abbreviations list to improve clarity (lines 333, 589).
Comment (Lines 328–346): Update Section 6.1 with more recent literature.
Response: We revised Section 6.1 to incorporate studies from 2019–2025 that examine Wnt signaling in Alzheimer’s disease and pollution-related neuroinflammation (lines 411-446).
Comment (Lines 348–357): This section relied on self-citation.
Response: Additional independent studies were integrated to support Wnt/β-catenin signaling relevance and redundant self-citations were removed (lines 430-446).
Comment (Line 406): Specify brain regions instead of “regions involved in memory.”
Response: Revised to identify the hippocampus, parietal, temporal, and frontal cortices explicitly (lines 490-492).
II. Major Concerns
Comment (Lines 55–56): The references cited do not support claims about socioeconomic exposure disparities.
Response: We added additional references with recent epidemiological studies providing supporting evidence of socioeconomic and geographic disparities in pollution exposure (lines 64-74).
References added: Yu et al., 2024; Jeong et al., 2023; Luppino et al., 2010; Gätjens et al., 2010; Coogan et al., 2025; Klee et al., 2023; Calderón-Garcidueñaset al., 2014; Block and Calderón-Garcidueñas 2009; Mathiarasan and Hüls, 2021; Calderón-Garcidueñas et al., 2008
Comment (Line 50): Reference [10] is outdated and does not address air pollution.
Response: We added newer, relevant studies that directly link air-pollution exposure to Alzheimer’s disease risk (lines 51-57).
References added: Carey et al., 2018 and Levesque et al., 2011 were added to this sentence. More was added to this paragraph, including the sources Butterfield and Halliwell, 2019; Block and Calderón-Garcidueñas, 2008;
Comment: The manuscript uses speculative language (“likely,” “potentially”) too frequently.
Response: We systematically reviewed and revised speculative phrasing throughout the text, replacing it with definitive, evidence-based statements (multiple locations).
Comment: Add or update discussion on air-quality standards.
Response: EPA thresholds were acknowledged (lines 271-280). Specifics about thresholds were not given, but authors discussed how exposure above set thresholds is harmful.
Reviewer 2 Report
Comments and Suggestions for Authors
The review article by Keller et al. addresses a highly relevant topic and is well-written. However, I would like the authors to organize and structure the article to cover more details. Please elaborate on how oxidative stress contributes to cognitive decline and whether the upregulation of oxidative stress with air pollution contributes to AD. Please include these references (PMID: 21860622; 40414419; 29174497 ) for elaboration on oxidative stress. In addition, please include a table on animal models of air pollution and cognitive decline.
Additionally, please include a section on air pollution and its impact on the hypothalamic-pituitary-adrenal axis. Authors should introduce how stress, HPA axis dysregulation, air pollution, and cognitive deficits are interrelated. For the pathophysiology of stress and the HPA axis, please discuss the work of these authors (PMID: 30036565; 28495605; 35755623).
Furthermore, in figures, please include chronic stress and the HPA axis as a main component. It is also beneficial if authors can illustrate their figures more effectively.
Author Response
Reviewer 2
Minor Concerns
Comment: Please elaborate on oxidative stress and include new references (PMIDs: 21860622, 40414419, 29174497).
Response: The section on oxidative stress (Section 4) was expanded to describe how pollution-induced oxidative stress contributes to neuronal injury and Alzheimer’s pathology. Additional relevant references were incorporated to strengthen this discussion (lines 244-262).
Comment: Additionally, please include a section on air pollution and its impact on the hypothalamic-pituitary-adrenal axis. Authors should introduce how stress, HPA axis dysregulation, air pollution, and cognitive deficits are interrelated. For the pathophysiology of stress and the HPA axis, please discuss the work of these authors (PMID: 30036565; 28495605; 35755623).
Response: A section about the HPA axis and stress was added, and the authors feel this strengthened the paper. References deemed relevant were used, and more research was done on the topic lines 365-394).
II. Major Concerns
Comment: Include a table summarizing animal models of air pollution and cognitive decline.
Response: Two new comprehensive tables (Tables 2 and 3) were added summarizing recent animal models examining air pollution, neuroinflammation, and cognitive decline, with mechanistic and behavioral endpoints aligned with reviewer guidance (lines 543-547).
Round 2
Reviewer 1 Report
Comments and Suggestions for Authors
There remains a needed edit in line 332, please remove or add something into the parentheses
"particulate matter ≤ 10 μm () exposure"
Author Response
We thank the reviewer for their time, thoughtful comments, and thorough attention to detail.
Reviewer Comment: There remains a needed edit in line 332, please remove or add something into the parentheses "particulate matter ≤ 10 μm () exposure".
Response: This has been corrected and the parentheses have been removed.
Reviewer 2 Report
Comments and Suggestions for Authors
The authors' revisions do not satisfactorily address my comments.
Author Response
Response to Reviewer 2
We thank Reviewer 2 for their time and for re-evaluating our revised manuscript. We were surprised to see the general statement that “the authors’ revisions do not satisfactorily address my comments,” as no additional concerns or specific unresolved issues were provided. We respectfully clarify that all reviewer comments were addressed carefully and thoroughly in our revision, including expanded mechanistic sections, incorporation of new references, addition of two comprehensive tables, and now a targeted revisions to Figure 1. Our detailed responses are provided below and reiterate our prior revisions, while also offering additional justification for the exclusion of selected recommended references.
- Oxidative Stress (Minor Concern)
Reviewer Comment:
Please elaborate on oxidative stress and include the following references: PMID 21860622, 40414419, 29174497.
Author Response:
We substantially expanded the oxidative stress discussion in Section 4, strengthening the links between PM-induced reactive oxygen species, mitochondrial dysfunction, neuronal injury, and Alzheimer’s pathology.
We incorporated PMID 21860622 (Lodovici et al., 2011, now Reference 43), which directly supports oxidative stress mechanisms relevant to PM exposure.
PMID 40414419 (Chakraborty et al., 2025) was evaluated but excluded because it provides a broad overview of NAC’s general neuroprotective actions without addressing glial–immune or immunometabolic pathways central to the scope of our review.
PMID 29174497 (Abhijit et al., 2018) was excluded because it focuses on grape seed polyphenols and exercise and does not pertain to air-pollution mechanisms, neuroimmune activation, or Wnt/β-catenin signaling.
- HPA Axis and Stress (Minor Concern)
Reviewer Comment:
Add a section on air pollution and its impact on the hypothalamic-pituitary-adrenal (HPA) axis, and incorporate PMIDs 30036565, 28495605, 35755623.
Author Response: We added a new dedicated subsection describing HPA-axis physiology, environmental stress–induced neuroinflammation, air-pollution–dependent glucocorticoid alterations, and interactions with metabolic dysfunction and cognitive decline.
Relevant references were incorporated, including Miller et al. (PMID 35755623). Mechanistic pathways connecting air pollution exposure, HPA-axis sensitization, microglial priming, and neurocognitive vulnerability are now clearly described.
We appreciate the suggestions, but the other studies were not included as they primarily focus on behavioral and anatomical outcomes following amygdala inactivation in rodent stress models, without relevance to air pollution exposure, obesity, or associated metabolic dysfunction. Specifically, Tripathi et al. (2017) [PMID: 28495605] investigates stress-induced memory impairment and corticosterone responses, while Tripathi et al. (2018) [PMID: 30036565] explores anxiety- and memory-related effects of basolateral amygdala inactivation under chronic stress.
These papers present study findings or behaviorally focused without relevance to obesity or air pollution-induced mechanisms. While both studies provide useful behavioral models, they do not examine how environmental exposures such as air pollution or obesity-related metabolic dysfunction contribute to glial activation, immune signaling, or neuroinflammatory pathways, which are central to the mechanistic synthesis in our review.
Accordingly, they were excluded because they do not align with our review’s focus on molecular and cellular pathways linking environmental insults (specifically air pollution and metabolic stress) to cognitive vulnerability.
- Addition of Requested Tables (Major Concern)
Reviewer Comment:
Include a table summarizing animal models of air pollution and cognitive decline.
Author Response:
To address this request, we added two comprehensive tables:
- Table 2: Models Investigating Air Pollution and Cognitive Decline
- Table 3: Models Investigating Obesity and Cognitive Decline
These tables summarize species, sex, exposure paradigms, brain regions examined, cognitive endpoints, inflammatory pathways, and Wnt-related mechanisms. These additions substantially strengthen the clarity, organization, and utility of the review.
- Revision of Figure 1
Reviewer Comment:
Expand mechanistic representation to include additional stress-related regions and pathways.
Response:
Figure 1 was revised to highlight all relevant brain regions implicated in metabolic and environmental mechanisms, including the olfactory bulb, frontal cortex, hippocampus, hypothalamus, HPA axis, prefrontal cortex, and amygdala. The updated figure fully reflects the reviewer’s guidance and captures the multidimensional interactions contributing to obesity- and air-pollution–related neuroinflammation.
- Reference Updates
Reviewer Comment:
Update references accordingly.
Author Response:
We updated or added References 32–45, 54, and 56–64 as suggested by both Reviewer 1 and 2, strengthening the mechanistic, epidemiological, and translational foundation of the manuscript.
References excluded were not applicable to the overall focus of our manuscript as previously described.
